# Enhancing Gelatine Hydrogel Robustness with Sacran-Aldehyde: A Natural Cross-Linker Approach

Maninder Singh [1,†], Alisha Debas [1,2,†], Gargi Joshi [1,3], Maiko Kaneko Okajima [1,4], Robin Rajan [1], Kazuaki Matsumura [1] and Tatsuo Kaneko [1,4,*]

1 Energy and Environment Area, Graduate School of Advanced Science and Technology, Japan Advanced Institute of Science and Technology, 1-1 Asahidai, Nomi 923-1292, Japan; msmanin22@gmail.com (M.S.); alisha.debas@monash.edu (A.D.); gargi.joshi@tu-dresden.de (G.J.); maiko@jiangnan.edu.cn (M.K.O.); robin@jaist.ac.jp (R.R.); mkazuaki@jaist.ac.jp (K.M.)
2 School of Chemistry, Monash University, Clayton, VIC 3800, Australia
3 B CUBE—Center for Molecular Bioengineering, Technische Universität Dresden, Tatzberg 41, 01307 Dresden, Germany
4 Key Laboratory of Synthetic and Biological Colloids, School of Chemical and Material Engineering, Jiangnan University, 1800 Lihu Avenue, Wuxi 214122, China
* Correspondence: tkaneko@jiangnan.edu.cn
† These authors contributed equally to this work.

**Abstract:** Tunable hydrogels have gained significant attention in the bioengineering field due to their designer preparation approach. Towards this end, gelatine stands out as a promising candidate owing to its desirable attributes, such as biocompatibility, ability to support cell adhesion and proliferation, biodegradability, and cost-effectiveness. This study presents the preparation of a robust gelatine hydrogel employing sacran aldehyde (SDA) as a natural cross-linker. The resulting SDA-cross-linked gelatine hydrogels (GSDA) display an optimal compressive stress of 0.15 MPa at 50% strain, five times higher than pure gelatine hydrogel. As SDA cross-linking concentration is increased, the swelling capacity of GSDA declines. This decline in swelling capacity, from 80% to 40%, is a result of strong crosslinking of gelatin with SDA. Probing further with FT-IR spectroscopy and SEM at the micron scale unveiled a dual-cross-linking mechanism within the hydrogels. This mechanism encompasses both short- and long-range covalent cross-linking, along with thermo-induced physical cross-linking, resulting in a significant enhancement of the load-bearing capacity of the fabricated hydrogels.

**Keywords:** gelatine; sacran; cross-linking; biocompatible; hydrogels; swelling





## 1. Introduction

Natural polymers are vital and appealing materials for numerous application fields, including the food industry, agriculture, biomedicine, and tissue engineering. With the growing demand for biomedical materials, there is increasing focus on customizing the structure, properties, and functions of natural polymers [1]. Gels are formed when hydrophilic polymer chains undergo chemical or physical cross-linking, resulting in an interconnected network [2–4]. Hydrogels, i.e., gels interspersed with water, have been extensively researched and applied due to their extended shelf life in the form of xerogels, water retention efficiency, and enhanced mechanical properties [5,6]. Hydrogels possess functional groups, such as $-NH_2$, $-SO_3H$, $-COOH$, $-OH$, and $-NHCOCH_3$, making them highly hydrophilic. Moreover, hydrogels exhibit properties akin to animal tissues due to these above-mentioned configurations [7–9].

Gelatine is a natural biopolymer containing functional groups, including amines, carboxylates, and hydroxyls. Its exceptional biodegradability and biocompatibility have made it a popular choice for a variety of biomedical applications, such as wound dressings, surgical treatments, and tissue engineering [10,11]. However, the practical use of gelatine

is limited by its stability only at lower temperatures (35–43 °C). Upon exposure to higher temperatures (49–60 °C), the secondary bonding structure tends to weaken, causing the physical network to break down, resulting in poor mechanical and thermal properties of gelatine hydrogels [12]. To overcome these limitations and expand the range of applications of gelatine, stabilization of its hydrogels is critical. This can be achieved through chemical modification or by blending gelatine with other biopolymers [13,14].

Sacran, a polysaccharide extracted from *Aphanothece sacrum*, is characterized by its multifunctional anionic chains. With an impressive molecular weight of up to $1.6 \times 10^7$ g/mol, sacran demonstrates super water absorbance characteristic [15]. Additionally, sacran exhibits valuable properties such as anti-inflammatory, anti-allergic, and wound-healing abilities, making it highly suitable for diverse biomedical applications [16–19]. Sacran has been demonstrated to form polyion complexes with collagen [20,21]. Furthermore, sacran has been blended with cellulose nanofiber (CNF-TEMPO) and Ag, bestowing a synergetic effect to curb bacterial and microbial infections [22,23]. The structural modification of sacran or it's blend with gelatine has never been investigated. This article deals with the novel approach of examining the performance of crosslinked gelatine gels containing dialdehyde moieties of sacran.

Polysaccharide dialdehyde [24] is one of the reactive derivatives of polysaccharides, obtained after periodate or TEMPO oxidation, which contains polyaldehyde structures, similar to other cross-linking agents like glutaraldehyde, alginate dialdehyde (ADA), dialdehyde cellulose (DAC), dextran dialdehyde (DDA), and oxidized xanthan gum (OXG) [25–30]. These active moieties in polysaccharide dialdehydes are capable of crosslinking with free amino groups in gelatine. Nevertheless, there has not been any research published on the application of SDA as a crosslinking agent.

A review article examined various natural polymers that have been explored for enhancing PVOH-based films used in food packaging. It specifically highlights starch, chitosan, cellulose, and gelatin, due to their low cost, renewability, abundance, sustainability, biocompatibility, and biodegradability. Additionally, this review briefly discussed the use of PVOH in conjunction with bio-waste-based films. Finally, it elaborates on the current research trends (2016–2021) regarding the combined use of PVOH-based and natural polymer films for food packaging applications [1].

This study aims to explore the potential use of sacran aldehyde (SDA) as a natural cross-linker in enhancing the stability of gelatine hydrogels. The cross-linking density, swelling properties, and surficial morphology of cross-linked gelatine hydrogels were analyzed and discussed using various characterization techniques. This cross-linked network is responsible for the improved rigidity and mechanical strength observed in the hydrogels. By utilizing sacran as a cross-linker and confirming the formation of sacran-aldehyde through FT-IR analysis, the study successfully demonstrated a strategy to enhance the stability of gelatine hydrogels. This advancement has significant implications for various biomedical applications, where stable and robust hydrogels are crucial for ensuring desired performance and functionality.

## 2. Materials and Methods

### 2.1. Materials

The gelatine from porcine skin was purchased from Sigma-Aldrich, Tokyo, Japan. Sacran ($M_w$ $1.6 \times 10^7$ g/mol) was obtained from Green Science Materials Inc. (Kumamoto, Japan). Sodium periodate was purchased from Nacalai Tesque, Kyoto, Japan. Ninhydrin (2,2-dihydroxy-1,3-indanedione) was purchased from TCI, Tokyo, Japan. All chemicals were used as received. A dialysis membrane was purchased from Funakoshi, Tokyo, Japan (MWCO 1 KDa; Φ 29 × 45 × 5 mm).

*2.2. Experimental*

2.2.1. Preparation of Sacran Aldehyde

Sacran (1 g) was dissolved in 0.1 M $H_2SO_4$ solution (50 mL) while continuously stirring at 40 °C for 4 h to obtain a homogeneous solution. Sodium periodate (4.2 mmol) was added to it, and the solution was stirred again at 40 °C for 4 h. The solution was cooled to room temperature and kept on stirring at 80 rpm overnight. The obtained product was then transferred into a cellulose dialysis membrane (MWCO: 1 KDa) and dialyzed in distilled water, changing the water every 12 h over 2 days. A cellulose membrane offers good chemical resistance, and low protein binding makes it an ideal candidate for various laboratory dialysis applications. The MWCO of 1 KDa was chosen on the basis of the molecular weight of the product to be dialyzed. This has to be lower than the molecular weight of the product. The dialyzed solution was freeze-dried, yielding 85% of the product. Sacran aldehyde molecular weight was calculated to be $6.8 \times 10^6$ g/mol from the SEC-MALLS technique.

2.2.2. Preparation of Gelatine Solution

Throughout this study, an aqueous solution was used to create hydrogels. To make the solution, gelatine granules (5 g) were dispersed in Milli-Q water (50 mL) and gently stirred for 20 min at 60 °C. The clear gelatine solution was stirred for another 5 min at 40 °C. and poured into a silicon mold to get pure gelatin clear hydrogel.

2.2.3. Syntheses of Cross-Linked Hydrogels

The gelatine solution was slowly added to freeze-dried sacran aldehyde with continuous stirring overnight to obtain sacran aldehyde cross-linked gelatine hydrogels. Four hydrogel samples of gelatine cross-linked with sacran aldehyde were prepared, as shown in Scheme 1.

**Scheme 1.** Formation of gelatine sacran aldehyde (GSDA) cross-linked hydrogels.

Sacran aldehyde cross-linked gelatine hydrogel samples of 1, 1.5, 2, and 3 wt.% were prepared by adding 100, 150, 200, and 300 mg of sacran aldehyde, respectively, to glass bottles containing 10 mL of gelatine solution (10 $w/v\%$). The mixture was then stirred at 40 °C for 4 h to complete the cross-linking reaction. The air bubbles formed during stirring were removed using centrifugation at 8000 rpm for 1 h. The resulting mixture was slowly poured into a silicon mold, avoiding the formation of air bubbles to give it a particular shape. The obtained gels became dense and darker in shade as the ratio of the crosslinker

was increased. All the gels were kept in a dry environment under controlled humidity (RH 30%, 25 °C) until further characterization.

### 2.2.4. Measurement of Mechanical Properties of Hydrogels

The mechanical properties of the pure gelatine and cross-linked GSDA samples were investigated by employing them in compression testing. Pure gelatine and cross-linked GSDA samples were kept in a humidity-controlled environment (relative humidity 30%, temperature 25 °C) for 2 days before being subjected to mechanical properties analysis. A total of six samples were subjected to mechanical testing. A compressing probe was set up on an Instron 3365 machine using a 5 kN load cell with a crosshead speed of 1 mm/min. The gels were cut into 5 mm × 5 mm × 5 mm cubic geometry for compression measurements. The measurement was repeated three times to calculate the error involved.

### 2.2.5. Measurement of Swelling Degree of Hydrogels

The swelling properties of the cross-linked hydrogels were studied by incubating the gels in Phosphate Buffer Saline (PBS pH 7) at room temperature. The gels were slightly blotted and weighed every 30 min until the gels started dissolving. The weight of the swelled gels was then compared with the weight of dried gels to evaluate the swelling ratio of hydrogels. The gel samples were weighed, and the swelling degree, $q$, was estimated as a weight ratio using the following equation,

$$q\% = \frac{W_s}{W_d} \times 100$$

$q$: swelling degree
$W_s$: weight of swelled gel
$W_d$: weight of dried crosslinked film

### 2.2.6. Scanning Electron Microscopy

The dried gel membranes were coated with Au using a magnetron sputtering system (MSP-IS, Vacuum Device, Shoreview, MN, USA) and observed under a desktop scanning electron microscope (Hitachi, TM3030plus, Tokyo, Japan) with an acceleration voltage of 15 kV using standard scanning mode.

### *2.3. Characterization of Pure Sacran, Sacran Aldehyde and Gelatine*

#### 2.3.1. Molecular Weight Estimation Using SEC-MALLS

The prepared sacran solution was kept at 4 °C prior to use. The absolute molecular weight ($M_w$) of sacran was determined using size exclusion chromatography combined with multi-angle static light scattering (SEC-MALLS) to be >$10^7$ g/mol [19]. Sacran has negative surface charges because of the presence of carboxylate anions and sulfate anions. The absolute molecular weight ($M_w$) of sacran aldehyde was measured to be between $10^6$–$10^7$ g/mol.

#### 2.3.2. Characterization of Cross-Linked Hydrogels

A ninhydrin (2,2-dihydroxy-1,3-indanedione) assay was conducted to determine the degree of cross-linking of the hydrogels. The lyophilized gels dissolved in 1 mL of distilled water were treated with 1 mL ninhydrin solution (1.5% *w/v* in ethanol). The mixture was heated for 1 h at 80 °C. After cooling down the mixture, color change was observed from orange to violet, representing effective cross-linking of the gelatine hydrogels and the optical absorbance was recorded using a UV-visible spectrometer (UV-1800, Shimadzu Corp., Kyoto, Japan) at a wavelength of 570 nm against a blank solution without gels.

Fourier Transform Infrared (FT-IR) spectra (Kyoto, Japan) were recorded with a PerkinElmer Spectrum One spectrometer between 4000 and 500 cm$^{-1}$ to confirm the formation of sacran-aldehyde.

## 3. Results and Discussion

### 3.1. Formation of Sacran Aldehyde

Sacran exhibits distinctive bands at 3328 cm$^{-1}$, CH$_2$ stretching vibration at 2925 cm$^{-1}$, polysaccharide (1→4) glycosidic bond stretching vibration at 1144 cm$^{-1}$, and C-O stretching vibrations at around 1007 cm$^{-1}$. In SDA, a new peak at 1732 cm$^{-1}$ reveals the presence of C=O stretching vibration in comparison to unmodified sacran (Figure 1 inset). Periodate can oxidize α-1,4-linked and α-1,6-linked anhydroglucoside units, forming either dialdehyde or aldehyde groups based on the chemical pathway, as reported by Bruneel et al. Also, a slight shoulder at 2726 cm$^{-1}$ for C-H stretching of aldehyde confirmed the formation of aldehyde functional groups. Additionally, a slightly red-shifted OH signal at 3328 cm$^{-1}$ suggests that aldehyde or dialdehyde groups were introduced into the sacran chain through oxidation.

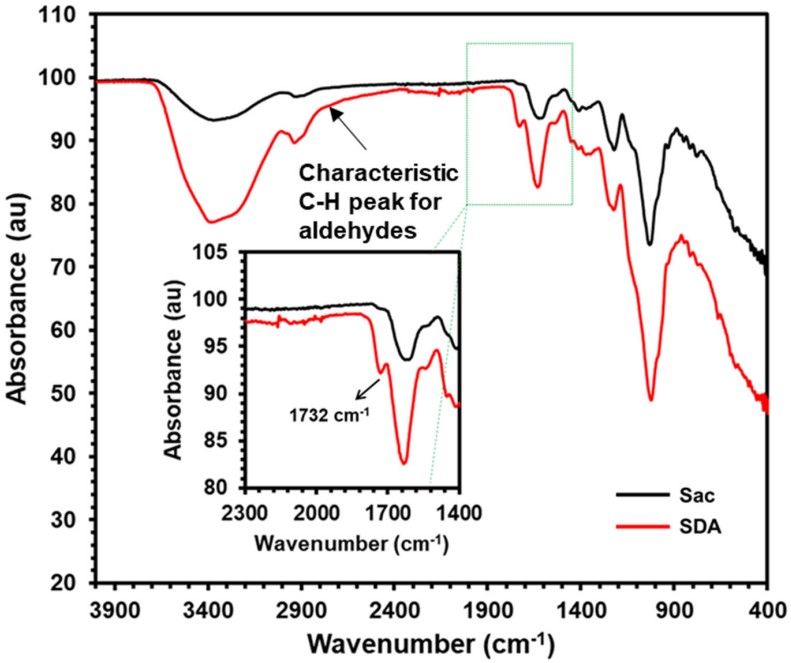

**Figure 1.** FT-IR spectra of Sacran (Sac) and Sacran aldehyde (SDA).

To improve the properties of gelatine hydrogels, sacran, a cyanobacterial polysaccharide, was employed. This was achieved by oxidizing sacran in the presence of sodium periodate, converting its hydroxyl groups to aldehyde [31]. To verify the successful formation of sacran aldehyde, the samples were tested using Fourier Transform Infrared (FT-IR) spectroscopy. A distinct peak at 1732 cm$^{-1}$ wavelength (Figure 1), indicating the oxidation of the hydroxyl groups and the formation of sacran aldehyde, was observed. Importantly, in the spectra of pure sacran, no peak was present at this specific wavelength, confirming the effectiveness of the oxidation process. Cross-linking occurred between these –CHO groups from sacran aldehyde (SDA) and the –NH$_2$ groups present in the lysine and hydroxylysine residues of gelatine (as illustrated in Scheme 1).

### 3.2. Cross-Linking of Hydrogels

Firstly, a clear gelatine solution was prepared by dissolving gelatine granules in distilled water and continuous stirred for 20 min at 60 °C followed by lowering temperature to 40 °C. The clear solution was poured into silicon mold to obtain clear gelatine hydrogel as shown in Figure 2a, Four different gelatine hydrogels cross-linked with sacran-dialdehyde were synthesized as shown in Figure 2b,c. The preparation process involved the gradual addition of gelatine solution to freeze-dried SDA under continuous stirring. The slow addition and continuous stirring allowed for effective mixing and reaction between gelatine and sacran-dialdehyde, ensuring the formation of covalent bonds along with the subsequent

establishment of a three-dimensional network structure. Moreover, the cross-linking density could be manipulated by adjusting the concentration of sacran-dialdehyde in the gelatine solution. As a result, hydrogels with varying degrees of cross-linking were obtained, each exhibiting distinct properties.

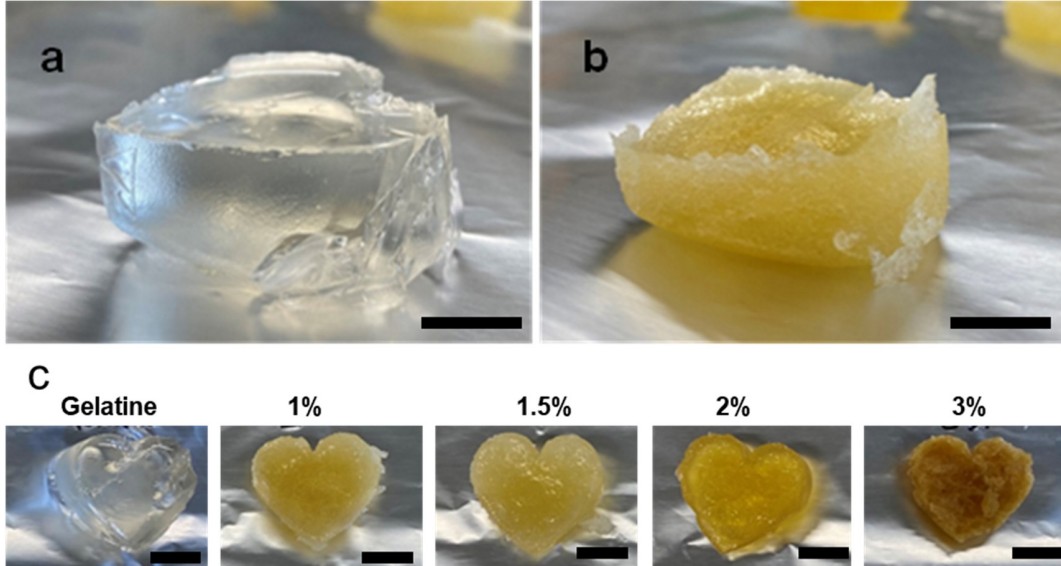

**Figure 2.** (**a**) Pure Gelatine; (**b**) Cross-linked gelatine (the color change represents effective cross-linking) (**c**) Gelatine hydrogels cross-linked with different amounts of sacran-dialdehyde. The scale bar in all images is 1 cm.

### 3.3. Effective Cross-Linking

To quantitatively assess the effective cross-linking in gelatine hydrogels, the ninhydrin test was utilized, which measures the presence of unreacted free amines. By employing the following equation, and from the results of UV-Vis spectroscopy of all the samples (as shown in Figure 3a), the extent of cross-linking within the hydrogel samples was evaluated.

$$\text{Degree of crosslinking }(\%) = \left\{1 - \left(\frac{\text{Absorbance of cross} - \text{linked gel}}{\text{Absorbance of noncross} - \text{linked gel}}\right)\right\} \times 100$$

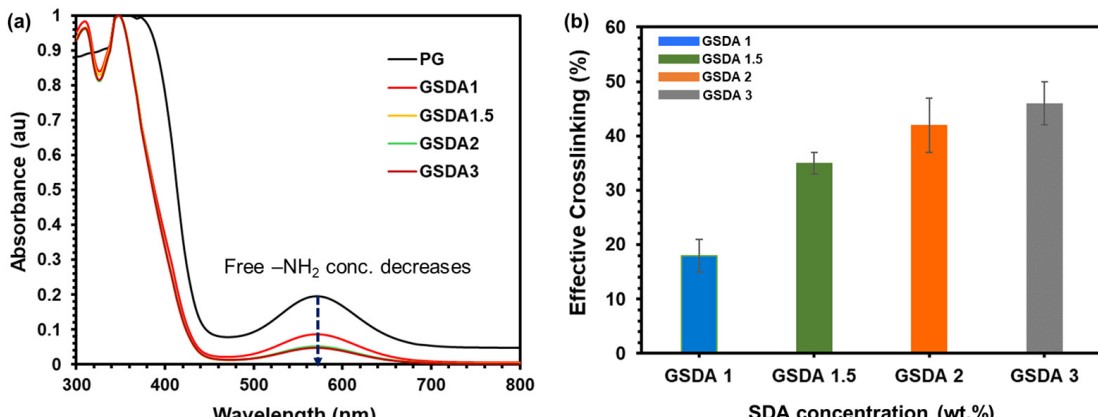

**Figure 3.** (**a**) UV-Vis data for the uncrosslinked and crosslinked samples. (**b**) Change of cross-linking degree of gelatine hydrogels as a function of sacran-dialdehyde amount. (arrow in (**a**) depicts the descending -NH$_2$ concentration).

A ninhydrin assay was used to measure the percentage of free amino groups in gelatin samples, which can be converted to the degree of crosslinking by comparing to

uncrosslinked gelatin. The content of free amino in the sample was directly proportional to the absorbance of the solution after being heated with ninhydrin. The degree of cross-linking exhibited a progressive trend, rising from 18% to 46% as the amount of oxidized sacran was elevated from GSDA 1 to GSDA 3 (as depicted in Figure 3b). As the content of aldehyde groups increased, more sites became available for the amino groups present in the gelatine molecules to react. Consequently, a greater number of covalent bonds were formed, leading to a denser and more interconnected network within the gelatine hydrogels [32].

The ninhydrin assay provided valuable insights into the quantitative assessment of cross-linking efficiency, offering a reliable method to determine the impact of varying cross-linker concentrations on the resulting gelatine hydrogel structures [33]. This information is crucial for tailoring the mechanical properties and stability of hydrogels to suit specific biomedical applications, where controlled and tunable cross-linking densities are of significant importance [34]. The cross-linking density ($\rho$) was calculated using the following equation.

$$\rho = Eq\frac{1}{3}/RT$$

where $R$ and $T$ are the gas constant and absolute temperature, respectively. The results are summarized in Table 1. The $NH_2$ content ($N$) was calculated [35] using the following equation and UV-Vis spectra.

$$N = (1 - \rho) \times N_{\text{non-crosslink}}$$

$$N_{\text{non-crosslink}} = N_g \times m_g$$

$$N_g = \frac{2 \times Abs \times 0.02}{1.46 \times 10^4 \times (b \times x)}$$

**Table 1.** Crosslinking density and $NH_2$ content in all samples.

| Samples | $\rho$ (mol/m$^3$) | Abs (au) | $N_{\text{non-crosslink}}$ ($10^{-6}$ mol/g) | $N$ ($10^{-7}$ mol/g) |
|---|---|---|---|---|
| PG | -- | 0.194 | 1.1 | -- |
| GSDA 1 | 0.001526619 | 0.08 | 0.44 | 4.38 |
| GSDA 1.5 | -- | 0.048 | 0.26 | -- |
| GSDA 2 | 0.00176189 | 0.041 | 0.22 | 2.24 |
| GSDA 3 | 0.001418368 | 0.039 | 0.21 | 2.13 |

In other words, a reversible covalent crosslinking bond was formed between the amino group of gelatin and the aldehyde group of SDA via the Schiff base reaction. Second, the hydroxyl groups of SDA and the amino groups on the surface of gelatin formed a hydrogen bond that produced physical crosslinks. The dynamic SDA hydrogel network was effectively created by double crosslinking using both chemical and physical methods (Scheme 1). The SDA hydrogel synthesis method has the advantages of moderate and fast reaction conditions, a simpler and more effective process, and the avoidance of the requirement for additional crosslinking agents or initiators [36].

### 3.4. Degree of Swelling

The swelling degree ($q$) of the hydrogels was determined using the following equation:

$$q = \left\{ \left( \frac{\text{Weight of swollen gel}}{\text{Weight of dry gel}} \right) - 1 \right\} \times 100$$

The experiment revealed that all the gels reached equilibrium swelling within 120 min. As depicted in Figure 4 and Table 2, there was a decrease in the swelling percentage as the content of oxidized sacran cross-linker increased, indicating more effective cross-linking in the gelatine hydrogels [37].

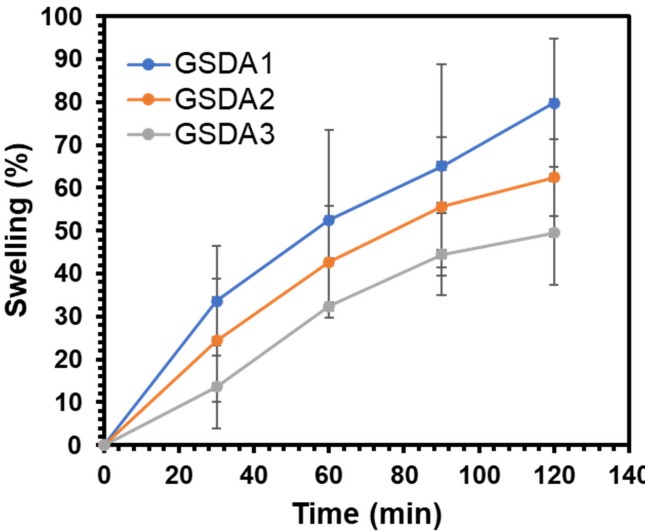

**Figure 4.** Time course of swelling degree change of sacran aldehyde cross-linked gelatine hydrogels.

**Table 2.** Mechanical, swelling and crosslinking behavior of the uncrosslinked and crosslinked samples.

| Samples | $\sigma$ (MPa) | $E$ (GPa) | $q$ (g/g) | $\rho$ ($10^{-3}$ mol/m$^3$) | $N$ ($10^{-7}$ mol/g) |
|---|---|---|---|---|---|
| PG | 0.04 | 1.17 | -- | --- | -- |
| GSDA1 | 0.06 | 1.65 | 0.70 | 1.53 | 4.38 |
| GSDA1.5 | 0.07 | 1.95 | -- | -- | -- |
| GSDA2 | 0.10 | 2.15 | 0.62 | 1.76 | 2.24 |
| GSDA 3 | 0.14 | 2.19 | 0.49 | 1.42 | 2.13 |

$\sigma$ and $E$ are the tensile strength and Young's Modulus. $q$ and $\rho$ are the swelling degree and crosslinking density.

### 3.5. Mechanical Properties

The enhancement of oxidized aldehyde groups for effective cross-linking in gel formation brings about notable improvements in the rigidity and stability of the resulting gels. These findings are depicted in Figure 5. The stress–strain curve shows that the incorporation and systematic increase in GSDA weight% resulted in increased stress with the same mechanical strain, as shown in Figure 5a. The mechanical strength of the gels correlates directly with the degree of cross-linking. This increase in mechanical strength makes the gels capable of withstanding higher applied pressures, particularly evident in the 50% compression tests. Cross-linking plays a crucial role in the mechanical properties of hydrogels, affecting their overall performance [34]. By introducing oxidized aldehyde groups through SDA, the gelatine matrix undergoes a structural transformation, leading to the formation of a robust and rigid network. This strengthened network is characterized by enhanced interactions between the polymer chains, resulting in a cohesive and stable gel structure. The resulting tensile strength with incorporated GSDA samples with varying wt.% is shown in Figure 5b and Table 2.

The newly formed bonds act as bridges, connecting neighboring gelatine molecules and reinforcing the gels' overall structure. During compression testing, the ability of the gels to resist deformation is significantly influenced by their cross-linking density. Hydrogels with higher cross-linking exhibit increased resistance to compression forces due to the enhanced intermolecular interactions. This property is valuable, in applications such as tissue engineering scaffolds, where the ability to withstand mechanical stresses is essential for the successful integration and support of cells.

However, it is essential to strike a balance in the degree of cross-linking to avoid potential drawbacks. Excessive cross-linking can lead to a reduction in the hydrogel's water uptake capacity, hindering its ability to absorb and retain moisture [38]. In certain biomedical applications, the hydrogels' ability to hold and release water or bioactive substances

is crucial for their efficacy. Therefore, optimization of the cross-linking density becomes a critical aspect in tailoring the properties of the hydrogels to suit specific applications.

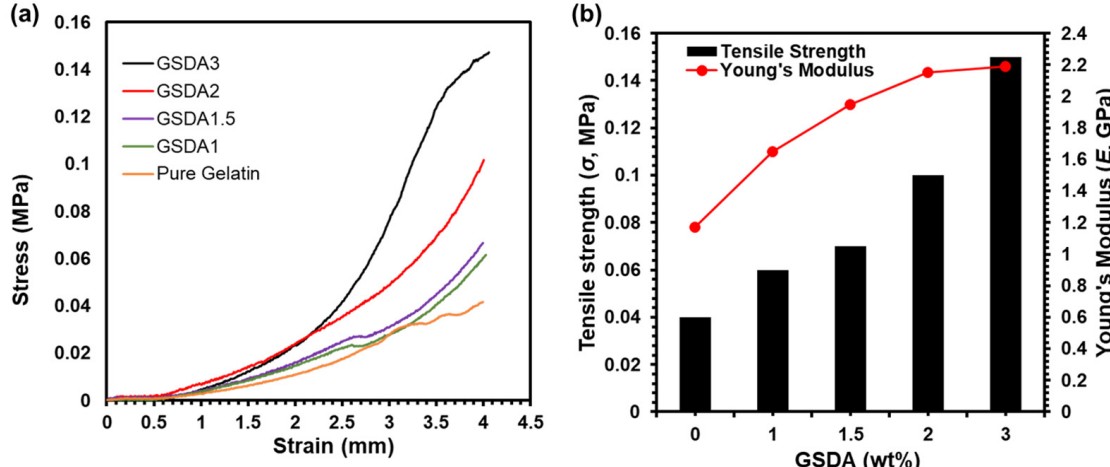

**Figure 5.** (**a**) Stress–Strain of gelatine hydrogels cross-linked with sacran-dialdehydes, (**b**) Mechanical data of gelatine hydrogels cross-linked with sacran-dialdehydes.

### 3.6. Morphology of Hydrogels

Samples of crosslinked hydrogels were freeze-dried before being subjected to SEM analysis. Freeze-drying kept the hydrogel structure of the samples intact [39]. SEM images of the hydrogels (Figure 6) reveal distinct differences in morphology between gelatine with and without oxidized polysaccharides. As cross-linking becomes more effective, the hydrogel's pore size decreases. On the other hand, pure gelatine exhibits a fibril-like morphology.

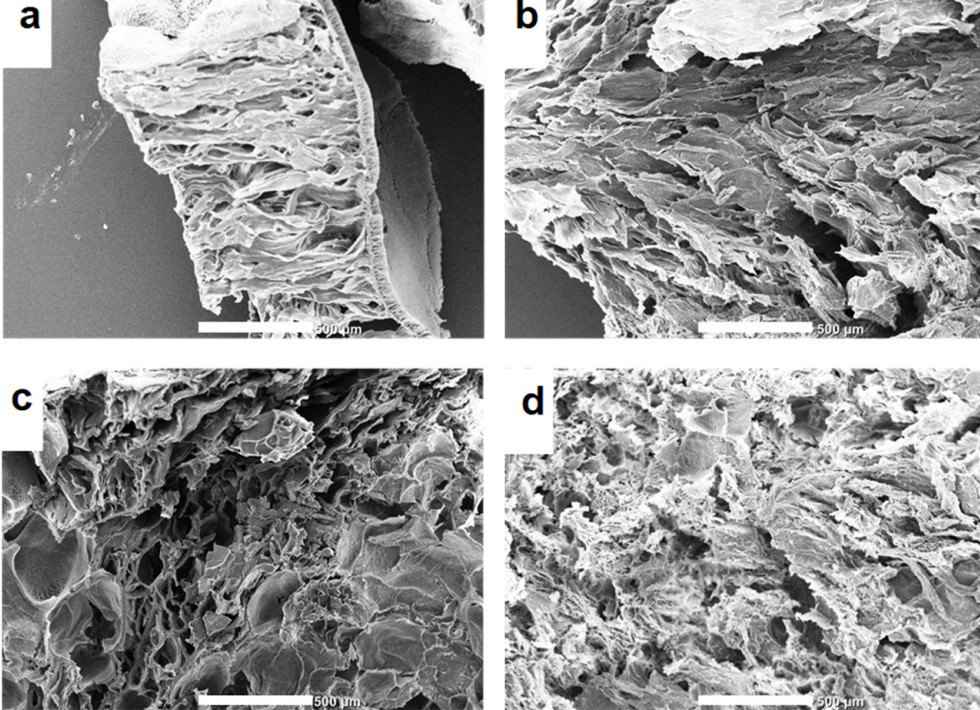

**Figure 6.** SEM images of (**a**) pure gelatine; (**b**) GSDA1; (**c**) GSDA2; (**d**) GSDA3. Scale bars in the images represent 500 μm.

The SEM images clearly illustrate how the introduction of oxidized polysaccharide as a cross-linker impacts the overall structure of the gelatine hydrogels. With increased cross-linking, the gelatine chains are more tightly connected, leading to a reduction in the size of the pores within the gel matrix. This decrease in pore size indicates a more compact and homogeneous network of the hydrogel, which can influence its physical and mechanical properties [40].

In contrast, the pure gelatine hydrogel exhibits a fibril-like appearance, characteristic of the native state. Without the presence of the cross-linker, the gelatine chains remain relatively unconnected.

## 4. Conclusions

Despite gelatine hydrogels having numerous biomedical applications such as wound healing, adhesives, plasma expanders, and drug delivery, their practical utility is limited due to poor stability at higher pressures and temperatures. This research aimed to address this limitation by synthesizing and characterizing gelatine hydrogels using a naturally occurring polysaccharide cross-linker called sacran, in the form of sacran aldehyde. By increasing the content of the aldehyde group for gelatine matrix preparation, effective cross-linking was achieved up to 45%, resulting in the formation of a rigid network within the gelatine hydrogels. This led to a reduction in the water uptake capacity of the hydrogels. Simultaneously, the degree of cross-linking was found to significantly enhance the mechanical stability of the gels, increasing it by five times compared to pure gelatine. The incorporation of sacran aldehyde as a cross-linker for gelatine opens up new possibilities and applications for these hydrogels in the field of biomedicine.

**Author Contributions:** Conceptualization, M.S., M.K.O. and T.K.; Data curation, M.S. and G.J.; Formal analysis, A.D. and G.J.; Funding acquisition, G.J. and T.K.; Investigation, M.S. and G.J.; Methodology, M.S., A.D., G.J., R.R. and T.K.; Project administration, T.K.; Supervision, T.K.; Validation, T.K.; Writing—original draft preparation, M.S., G.J. and T.K.; Writing—review and editing, M.S., M.K.O., R.R., K.M. and T.K. All authors have read and agreed to the published version of the manuscript.

**Funding:** The authors are thankful to the financial support provided by a Grant-in-aid, A-step (AS2915173U) of JST. G.J. is grateful for the Research Fellowship from the Japan Society for the Promotion of Sciences (JSPS) and the JSPS KAKEHNI Grant number JP18J11881.

**Institutional Review Board Statement:** Not applicable.

**Data Availability Statement:** Data are contained within the article.

**Conflicts of Interest:** The authors declare the following financial interests/personal relationships which may be considered as potential competing interests: Tatsuo Kaneko reports financial support was provided by JST Adaptable and SeamLess Technology Transfer Program Through Target-driven R and D (A-STEP, AS2915173U). Gargi Joshi reports financial support was provided by the Japan Society for the Promotion of Science (KAKEHNI, JP18J11881).

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
