# Peer review of "Enhancing Gelatine Hydrogel Robustness with Sacran-Aldehyde: A Natural Cross-Linker Approach"

_2673-4176, doi:10.3390/polysaccharides5030021_

Round 1

Reviewer 1 Report

Comments and Suggestions for Authors

Authors synthesize Gelatine Hydrogel Robustness using natural cross-linkers. This manuscript is well organized. It can be published after revision. My following comments are below:

The abstract needs to be modified critically, such as why swelling is enhanced. Mention the reason. Insert some numerical values.

Line 58-61, need to rewrite more clearly at present limited information. Here authors need to highlight the research gap by considering crosslinking.

All references in the text after a full stop. It should be before full stop.

In the section 2, please make a section as materials.

Line 87, why authors use cellulose dialysis membrane for the MWCO: 1KDa? Any reason? Why not high mol. Wt. or low mol. Wt.?

Line 94, ml as mL. correct these errors throughout the manuscript.

Line 131, equation should 100 as it will be converted to %.

Present the cross-linking density? Which is imp for this paper and its effect.

Figure 1, y axis should absorbance (a.u.)? remover scale thicks.

Line 96, what is picture a? Line 103, sentence should not start with a number?

All discussion are in surface, need to improve?

In the introduction, briefly explain why authors use natural cross-linkers than other crosslinkers. and compare some published data. Authors can indicate this reference in the introduction section, Food Packaging and Shelf Life 33 (2022) 100904

Scheme 1 text is not visible. Improve it.

In Fig 1. the absorbance unit is missing. Y axis should be a.u. unit. Please remove axis thik and design the figure.

Explain more details in the crosslinking section results with the mechanism. Please provide a schematic diagram.  Authors can follow this for the mechanism Progress in Organic Coatings 183 (2023) 107749 and indicated in the manuscript.

The conclusion should be one paragraph. Avoid general summary.

Comments on the Quality of English Language

Minor checking needed.

Author Response

Authors synthesize Gelatine Hydrogel Robustness using natural cross-linkers. This manuscript is well organized. It can be published after revision. My following comments are below:

  1. The abstract needs to be modified critically, such as why swelling is enhanced. Mention the reason. Insert some numerical values.

We have revised the abstract and added following text on page 1 Line 25-26,

“This decline in swelling capacity is result of strong crosslinking of gelatin with SDA. Swelling degree decreases from 80 to 40%.”

  1. Line 58-61, need to rewrite more clearly at present limited information. Here authors need to highlight the research gap by considering crosslinking.

We have added following text on page 2 Line 71-73,

“These active moieties in polysaccharide dialdehydes are capable of crosslinking with free amino groups in the gelatine. Nevertheless, there has not been much research published on the application of SDA as a crosslinking agent to stabilize gelatin hydrogels.”

  1. All references in the text after a full stop. It should be before full stop.

We have made changes as requested.

  1. In the section 2, please make a section as materials.

We have made changes as requested.

  1. Line 87, why authors use cellulose dialysis membrane for the MWCO: 1KDa? Any reason? Why not high mol. Wt. or low mol. Wt.?

Thank you for your comment.

Cellulose dialysis membranes offer low protein binding, good chemical resistance, and affordability, making them ideal for general laboratory use and a variety of dialysis applications.

The MWCO for dialysis membranes in gelatine-SDA material is determined by the its molecular weight. Choose a MWCO that is slightly lower than the molecular weight of the material to retain it while allowing smaller contaminants to flow through. This ensures effective separation and purification. We have added following text on page 3 Line 110-113,

Cellulose membrane offers good chemical resistance; low protein binding makes it ideal candidate for various laboratory dialysis applications. The MWCO 1KDa are chosen on the basis of molecular weight of the product to be dialyzed. This has to be lower than the molecular weight of the product.”

  1. Line 94, ml as mL. correct these errors throughout the manuscript.

We have corrected

  1. Line 131, equation should 100 as it will be converted to %.

We have corrected the equation.

  1. Present the cross-linking density? Which is imp for this paper and its effect.

We have provided the requested data in the manuscript.

  1. Figure 1, y axis should absorbance (a.u.)? remover scale thicks.

We have corrected the figure.

  1. Line 96, what is picture a? Line 103, sentence should not start with a number?

Picture “a” is digital image of a gelatine hydrogel shown in line 199. We have changed the caption from picture to figure 2

We have corrected the sentence and highlighted it in line 128

Sacran aldehyde cross-linked gelatine hydrogels samples of 1, 1.5, 2, 3 wt. %.....”

  1. All discussion are in surface, need to improve?

We have improved the discussion section

  1. In the introduction, briefly explain why authors use natural cross-linkers than other crosslinkers. and compare some published data. Authors can indicate this reference in the introduction section, Food Packaging and Shelf Life 33 (2022) 100904

Following test has been added to the introduction in Line 35 and 75

Natural polymers are vital and appealing materials for numerous application fields, including the food industry, agriculture, biomedicine, and tissue engineering. With the growing demand for biomedical materials, there is increasing focus on customizing the structure, properties, and function of natural polymers.”

A review article examined various natural polymers that have been explored for enhancing PVOH-based films used in food packaging. It specifically highlights starch, chitosan, cellulose, and gelatin, due to their low cost, renewability, abundance, sustainability, biocompatibility, and biodegradability. Additionally, this review briefly discussed the use of PVOH in conjunction with bio-waste-based films. Finally, it elaborates on the current research trends (2016–2021) regarding the combined use of PVOH-based and natural polymer films for food packaging applications.”

  1. Scheme 1 text is not visible. Improve it.

We have revised scheme 1.

  1. In Fig 1. the absorbance unit is missing. Y axis should be a.u. unit. Please remove axis thik and design the figure.

We have redesigned the Fig according to reviewers comment.

  1. Explain more details in the crosslinking section results with the mechanism. Please provide a schematic diagram.  Authors can follow this for the mechanism Progress in Organic Coatings 183 (2023) 107749 and indicated in the manuscript.

We have added following explanation in Line 277 and scheme 1

In other words, a reversible covalent crosslinking bond was formed between the amino group of gelatin and the aldehyde group of SDA via the Schiff base reaction. Second, the hydroxyl groups of SDA and the amino groups on the surface of the gelatin formed a hydrogen bond that produced physical crosslinks. The dynamic SDA hydrogel network was effectively created by double crosslinking using both chemical and physical methods (Scheme 1). The SDA hydrogel synthesis method has the advantages of moderate and fast reaction conditions, a simpler and more effective process, and the avoidance of the requirement for additional crosslinking agents or initiators.”

  1. The conclusion should be one paragraph. Avoid general summary.

We have confined the text into a single paragraph avoiding generalization.

Reviewer 2 Report

Comments and Suggestions for Authors

Comments

The manuscript entitled “Enhancing Gelatine Hydrogel Robustness with Sacran-Aldehyde: A Natural Cross-Linker Approach” describes the incorporation of sacran aldehyde as a cross-linker for gelatine for the synthesis of gelatin hydrogels to improve the mechanical properties.

However there are major corrections which are required to be addressed.

1. Line 166, Page 4: Did the characteristic aldehyde peak was observed in IR for C-H aldehyde group between 2720-2740 cm-1? This will also confirm the conversion to Sacran aldehyde. If the peak is not observed please provide the reason.

2. Line 193-195 Page 6: Please provide the UV data, which measures the presence of unreacted free amine. 

Decision: Accept after suggested major revision.

Author Response

The manuscript entitled “Enhancing Gelatine Hydrogel Robustness with Sacran-Aldehyde: A Natural Cross-Linker Approach” describes the incorporation of sacran aldehyde as a cross-linker for gelatinefor the synthesis of gelatin hydrogels to improve the mechanical properties. However there are major corrections which are required to be addressed.

  1. Line 166, Page 4: Did the characteristic aldehyde peak was observed in IR for C-H aldehyde groupbetween 2720-2740 cm-1? This will also confirm the conversion to Sacran aldehyde. If the peak is not observed please provide the reason.

We have added the explanation in the results and discussion section of the manuscript.

  1. Line 193-195 Page 6: Please provide the UV data, which measures the presence of unreacted freeamine.

We have added UV data in the manuscript.

Decision: Accept after suggested major revision

Reviewer 3 Report

Comments and Suggestions for Authors

This study is devoted to the formation of imine crosslinked gelatin. The idea is not novel, gelatin was previously crossinked by different aldehyde containg polysaccharides including cellulose, dextran and xanthan gum derivatives, as well as by glutaraldehyde. Thus, please specify the novelty of this study. Also the rheological parameters such as storage modulus G', loss modulus G'' and tandelta are required to be provided to confirm gel structure (G'>G''). Overall, this work could be reconsidered after major revision. Some other comments are listed below:

- Line 37, please add "functional" before word "groups

- Line 96, please replace ref to "picture a" to relevant scheme or figure (in the articles there are figures or scheme and coudn't be  "picture")

- scheme 1 has too small font and unreadable

- please provide the concentration of NH2 groups in gelatin. You an refer to a previsouly reported papers (10.1073/pnas.222075512)

- please provide all important characteristic bonds in FTIR in fig. 1, i.e. OH, NH, C=O, C=N, C-O and CH

- please specify how you evaluate real crosslinks to get data on fig. 2, please, add error bars

- fig. 4 is raw mechanical test data. Please fit them to get Young's Modulus and tensile strength

- please specify storage and loss modulus of hydrogels with different crosslinking densities

- please comment how samples were dried for getting SEM images in fig. 5. It's known that ambient drying lead to collapse of hydrogel structure (10.1016/j.jtice.2020.03.004)

Comments on the Quality of English Language

Minor editing of English language required

Author Response

This study is devoted to the formation of imine crosslinked gelatin. The idea is not novel, gelatin was previously crossinked by different aldehyde containg polysaccharides including cellulose, dextran and xanthan gum derivatives, as well as by glutaraldehyde.

Thus, please specify the novelty of this study.

Novelty of this study has been added in the Line 63

 Also the rheological parameters such as storage modulus G', loss modulus G'' and tandelta are required to be provided to confirm gel structure (G'>G''). Overall, this work could be reconsidered after major revision. Some other comments are listed below:

This has been answered in question 8.

  1. Line 37, please add "functional" before word "groups

We have added the word “ functional” as requested

  1. Line 96, please replace ref to "picture a" to relevant scheme or figure (in the articles there are figures or scheme and couldn’t ibe  "picture")

We have changed as requested. We have changed the caption from picture to figure 2

  1. scheme 1 has too small font and unreadable

We have revised the figure

  1. please provide the concentration of NH2 groups in gelatin. You can refer to a previously reported papers (10.1073/pnas.222075512)

We have provided the UV data and NH2 concentration in Gelatin. We could not find the mentioned paper. We have referred another paper for the same.

  1. please provide all important characteristic bonds in FTIR in fig. 1, i.e. OH, NH, C=O, C=N, C-O and CH

We have provided all the characteristic peaks in FTIR with explanation.

  1. please specify how you evaluate real crosslinks to get data on fig. 2, please, add error bars

We have added error bars to the figure. We have added following explanation in the manuscript in Line 241, page 7

A ninhydrin assay was used to measure the percentage of free amino groups in gelatin samples, which can be converted to the degree of crosslinking by comparing to uncrosslinked gelatin. The content of free amino in the sample was directly proportional to the absorbance of the solution after being heated with ninhydrin.”

  1. 4 is raw mechanical test data. Please fit them to get Young's Modulus and tensile strength

We have provided requested mechanical data and renamed it as Fig. 5

  1. please specify storage and loss modulus of hydrogels with different crosslinking densities

We fully acknowledge the importance of the suggested data in your comment. Unfortunately, due to explain the time constraints and instrument unavailability, we are unable to conduct the requested changes within the current revision period. However, we believe that our current findings still provide significant insights through crosslinking densities.

We will definitely take your valuable suggestion into account for our future work, where we plan to conduct a detailed study of various crosslinked gels for the desired applications. We are confident that incorporating your recommendations in our subsequent studies will further validate and expand our current findings.

  1. please comment how samples were dried for getting SEM images in fig. 5. It's known that ambient drying lead to collapse of hydrogel structure (10.1016/j.jtice.2020.03.004

Following statement was added in line 336

Samples of crosslinked hydrogels were freeze dried before subjecting them to the SEM analysis. Freeze drying kept the hydrogel structure of the samples intact.”

Round 2

Reviewer 1 Report

Comments and Suggestions for Authors

The authors improved the manuscript 

Reviewer 2 Report

Comments and Suggestions for Authors

Decision: Accept in present form.

The revisions suggested have been addressed by the author.

Reviewer 3 Report

Comments and Suggestions for Authors

Authors have addressed all reviewers' comments. Paper could be published in present form

Comments on the Quality of English Language

Minor editing of English is required